# Multifield-Controlled Terahertz Hybrid Metasurface for Switches and Logic Operations

**DOI:** 10.3390/nano12213765

**Published:** 2022-10-26

**Authors:** Xilai Zhao, Yanan Jiao, Jiangang Liang, Jing Lou, Jing Zhang, Jiawen Lv, Xiaohui Du, Lian Shen, Bin Zheng, Tong Cai

**Affiliations:** 1Air and Missile Defense College, Air Force Engineering University, Xi’an 710051, China; 2Department of General Surgery, First Medical Center, Chinese People’s Liberation Army (PLA) General Hospital, Beijing 100024, China; 3China Nuclear Engineering Consulting Corporation, Beijing 100024, China; 4Interdisciplinary Center for Quantum Information, State Key Laboratory of Modern Optical Instrumentation, Zhejiang University Hangzhou Global Scientific and Technological Innovation Center, Zhejiang University, Hangzhou 310027, China

**Keywords:** terahertz hybrid metasurface, multifield control, multiband switching, logic gate operation

## Abstract

Terahertz (THz) meta-devices are considered to be a promising framework for constructing integrated photonic circuitry, which is significant for processing the upsurge of data brought about by next-generation telecommunications. However, present active metasurfaces are typically restricted by a single external driving field, a single modulated frequency, fixed switching speed, and deficiency in logical operation functions which prevents devices from further practical applications. Here, to overcome these limitations, we propose a hybrid THz metasurface consisting of vanadium dioxide (VO_2_) and germanium (Ge) that enables electrical and optical tuning methods individually or simultaneously and theoretically investigate its performance. Each of the two materials is arranged in the meta-atom to dominate the resonance strength of toroidal or magnetic dipoles. Controlled by either or both of the external excitations, the device can switch on or off at four different frequencies, possessing two temporal degrees of freedom in terms of manipulation when considering the nonvolatility of VO_2_ and ultrafast photogenerated carriers of Ge. Furthermore, the “AND” and “OR” logic operations are respectively achieved at two adjacent frequency bands by weighing normalized transmission amplitude. This work may provide an auspicious paradigm of THz components, such as dynamic filters, multiband switches, and logical modulators, potentially promoting the design and implementation of multifunctional electro-optical devices in future THz computing and communication.

## 1. Introduction

The terahertz (THz) spectral region, bridging the infrared and microwave band at 0.1–1 THz, has become one of the candidate spectra for upcoming prospective next-generation communication technology owing to its advantages of superior penetration, outstanding security and confidentiality, and high data rate with low latency [1,2,3,4,5]. To handle the imminent increase in wireless data traffic brought about by further development of societal informatization and intellectualization, there is an urgent requirement for efficient photonics and electronics devices capable of dynamically manipulating THz signals [5,6,7,8]. However, most of the natural materials are not suitable for direct THz wave modulation, which seriously hinders the applications of THz devices. Metasurfaces, favored for their customizable abilities regarding the tailoring of the amplitude, phase, and polarization of electromagnetic waves through artificially elaborated two-dimensional structures, are considered to be an effective avenue for compensating for the portability, flexibility, and reliability generally lacking in conventional materials [9,10,11,12,13]. Furthermore, to accommodate multiple complicated utilization situations, present metasurface-based devices place more emphasis on obtaining nonlinear, memory, and reconfiguration properties via hybridizing function matters [14]. Regarding these matters, active materials (such as semiconductors [15,16,17,18,19], phase change materials [20], 2D materials [21,22,23], Weyl semimetals [24], perovskites [25,26,27], and so on) have contributed significantly to establishing tuned meta-atoms with variously exceptional properties. Meanwhile, depending on the constructed units, a series of external stimuli excitations, including optical [28,29,30], electrical [31,32,33,34], thermal [35], and mechanical [36,37,38] approaches, have been jointly employed to promote active metasurfaces that are more multidimensional. Nowadays, metasurface components with active materials have an extensive range of promising applications, which extends a brilliant route toward actualizing integrated devices with high performance. 

Despite these significant achievements, there are still some limitations that preclude active metasurfaces from completely fulfilling their potential. To obtain ultrafast temporal manipulation of THz waves, semiconductors such as those made of silicon (Si), germanium (Ge), and gallium arsenide (GaAs) are preferred for the rapid carrier concentration alterations induced by optical pump irradiation [39]. However, some devices employing semiconductors often suffer from a single switching function at one frequency point or a very narrow band due to the inherent resonance of unit cells. These reactions are generally volatile and adscititious energy is required during the entire operation to maintain switch states, resulting in high energy consumption. Fortunately, the adoption of phase change materials (PCMs), including liquid crystals [40,41], germanium−antimony−telluride (GST) alloys [42,43], and vanadium dioxide (VO_2_) [35,44,45,46,47,48], offers a step toward solving such problems. After excitation by external stimuli, VO_2_ can transform between the insulator and metal phase over a longer period of time, thus its application can be beneficial in the construction of memory metasurfaces [49]. Furthermore, much research has demonstrated that VO_2_ can be subject to multiple external excitations, which greatly enhances flexibility and utility when managing interacted electromagnetic fields [46,50]. So far, the integration of meta-atoms combining semiconductors and VO_2_ to take full advantage of their properties has opened an opportunity for future THz active metasurfaces, which paves the way for the realization of diversified tuning approaches and spatiotemporal modulation [49,51]. However, the effect excited by various stimuli is always designed to only be a linear superposition. Nevertheless, less attention has been paid to equipping devices with the logical operation function, which is a vital ability when processing large amounts of data and complex information. Devices based on several active materials with multifield-controlled, multifunction, and digital processing capabilities remain elusive but exigent.

In this work, we propose a THz metasurface by combining of VO_2_ and Ge, which can be controlled by electrical and optical methods either individually or simultaneously. Acting on the unit structures with a coupling effect, the two material components dominate the strength of the toroidal dipole or magnetic dipole resonance, respectively. The influences of tuning approaches are imitated via the changes in material electrical conductivity, and the corresponding properties of the device are investigated. After excitation by voltage, the device can separately switch for relatively long periods nearby 0.362 and 0.483 THz thanks to the VO_2_ phase transition. In terms of the VO_2_ bridge phases, the device respectively switches to an ultrafast speed at 0.565 and 0.585 THz as Ge layers are pumped by the optical pulse laser. More importantly, the interactions generated by the two stimuli provide the logical functions of “AND” and “OR” gates in the range of 0.56–0.57 THz and 0.58–0.59 THz, respectively, by weighing normalized wave transmission amplitudes. These properties, rendered by combining tunable materials with THz metasurfaces, show the device has potential for application in fields such as optoelectrical switching, THz communication, and photonics computing.

## 2. Materials and Methods

Figure 1a demonstrates the working principle of the electrically and optically controlled hybrid metasurface that consists of four materials (gold (Au), VO_2_, Ge, and sapphire). Considering the excellent chemical stability and electrical conductivity, Au was utilized to constitute electrodes and metal parts of resonant units. Four gold split-ring resonators (SRRs) were located symmetrically on the upper surface, with the middle two called SRR_I_ and SRR_II_ and the outer two called SRR_III_ and SRR_IV_. In the common long side of SRR_I_ and SRR_II_, an embedded VO_2_ bridge was precisely connected to the electrodes and resonators, which could undergo a reversible phase transition triggered by the heat generated from external currents. The dielectric permittivity of VO_2_ can be described using a Drude model [51]: (1)$$\varepsilon \left(\omega \right){=\varepsilon }_{\infty }\;-\;\frac{\omega_{p}^{2}\left(\sigma \right)}{\omega^{2}+i\gamma \omega }$$
where $\varepsilon_{\infty }$ and γ are usually assumed to be 12 and 5.75 × 10^13^ rad/s, plasma frequency $\omega_{p}^{2}\left(\sigma \right)$ is dependent on conductivity, and σ is proportional to the free carrier density [44,52]. Additionally, to properly manipulate resonance intensity, semiconductor layers made of Ge were selected to wrap SRR_III_ and SRR_IV_. The entire structure was constructed based on a sapphire substrate with excellent light transmittance. Figure 1b,c illustrate the single unit cell and its geometrical dimensions, while the detail parameters which stem from an overall consideration of the bandwidth, operating frequency, modulation depth, sensitivity, and stability of the device are listed in Table 1. The fabrication process of the device is discussed in Appendix A.

The proposed metasurface’s performance was numerically investigated based on the time-domain solver of the full-wave electromagnetic software CST Microwave Studio 2019. Periodical boundaries were assumed to X and Y directions, and open boundaries were applied in the Z direction. The linear plane waves are normally incident with Y polarization along the negative Z direction. To imitate the phase transition of VO_2_ induced by electrical voltage, conductivity was varied from 2 × 10^2^ to 2 × 10^5^ S/m [50,52,53]. Similarly, the conductivity of Ge was varied from 1 to 4 × 10^3^ S/m to simulate the influence of optical pump irradiation [49,54]. It is essential to mention that a 400 nm optical pump (3.1 eV) is typically used to activate the photoinduced carriers in Ge (0.66 eV), and the pump frequency enables the Ge conductivity change to 4 × 10^3^ S/m that only leads to a slight alteration in undoped VO_2_ conductivity, thus ensuring independence between the two modulation approaches [52,55,56,57]. The normalized transmission spectra were calculated via the formula: |T(ω)| = |E_s_(ω)/E_r_(ω)| with E_s_(ω) and E_r_(ω) representing the amplitude of THz waves transmitted through the device and the thickness of bare sapphire, respectively [58].

Figure 2a indicates the behaviors of the current directions and magnetic field (H-field) when VO_2_ bridges are in the metal phase and Ge layers are unexcited, at which time strong resonances occur at the SRRs. Transmission spectra of the proposed metasurface for two materials with or without excitation are plotted in Figure 2b. To examine the working mechanism of the proposed hybrid metasurface, we separately simulated the two parts of the units, and the surface currents are shown in Figure 2c,e. Considering the nonvolatility of VO_2_, currents can flow through metal phase VO_2_ bridges after excitation for a certain period, resulting in opposite current loops on SRR_I_ and SRR_II_ surfaces. Such a current distribution motivates a circular closure of the oscillating H-field that rotates around the two SRRs, which characterizes the appearance of the toroidal dipole and vector T in the space [16]. In the absence of optical pump influence, the conductivity of Ge is approximately equal to 1 S/m, and the current distribution on the surfaces of SRR_III_ and SRR_IV_ manifests an obvious magnetic dipole type. Figure 2d,f show the simulated transmission spectra for the two different types of resonance modulation. Due to the influence of contrary direction currents flowing on Au electrodes, the resonance intensity of the toroidal dipole is weaker in terms of transmission dip value. 

## 3. Results and Discussions

### 3.1. Simulation of Electrically Tuned VO_2_ Bridges

We firstly investigated the electrical tuning properties of the hybrid metasurface by simply varying the conductivity of VO_2_ bridges in the simulation, and the normalized transmission spectra without or with the optical pump irradiation are depicted in Figure 3a,j, respectively. In the absence of external stimuli, Figure 3a presents a transmission dip of 0.02 located at 0.362 THz, accompanied by a peak of 0.95 appearing at 0.518 THz. With the increase in VO_2_ conductivity, the transmission valley constantly grew while the peak kept dropping. By fixing the VO_2_ conductivity at 2 × 10^5^ S/m to imitate the ultimate outcome achieved by electrical stimulation, the transmission amplitude at 0.362 THz reached a maximum of 0.81. Meanwhile, a new dip valued at 0.26 occurred at 0.481 THz. Similarly, supposing Ge conductivity of 4 × 10^3^ S/m, Figure 3j displays transmission amplitudes rising from 0.03 to 0.80 at 0.362 THz, and a peak of 0.86 at 0.508 THz is replaced by the new dip of 0.33 at 0.483 THz alongside the phase transition of VO_2_. To explain the modifications to transmission spectra, we modeled the alterations to the H-field transmitted through the surface by simply enhancing the VO_2_ conductivity and fixing the Ge conductivity to 1 S/m at 0.362 THz and 4 × 10^3^ S/m at 0.483 THz, as shown in Figure 3b–i. As observed, the dip nearby 0.362 THz is caused by the strong resonance on both sides of the gold electrodes, where the H-field gathers when VO_2_ is in the insulator phase. Increasing the conductivity of VO_2_ bridges gradually permitted the flux of currents, thus resulting in a toroidal resonance that affected SRR_I_ and SRR_II_. This was followed by a certain coupling resonance with SRR_III_ and SRR_IV_, signified by the transmitted H-field inside. Finally, as VO_2_ conductivity was set to 2 × 10^5^ S/m, the magnetic field surrounding the electrodes tended to vanish, and a toroidal resonance impacted SRR_I_ and SRR_II_ thereby leading to a transmission dip near 0.483 THz.

To quantitatively analyze the effect of electrical tuning, we adopted modulation depth (MD) to measure the switching ability of the device when dynamically manipulating the transmission of THz waves. For the purpose of precisely unveiling the conversion degree, MD is defined as:(2)$$\mathrm{MD}=\left|\frac{T_{\mathrm{on}}{\;-\;T}_{\mathrm{off}}}{T_{\mathrm{on}}}\right|\;\times \;100\%$$
where *T*_on_ or *T*_off_ represents the transmission amplitude when the device is on or off, respectively. By fixing the conductivity of Ge layers at 1 S/m, Figure 4a,b present MD as a function of frequency in the ranges of 0.30–0.40 THz and 0.45–0.53 THz for varying VO_2_ conductivity. Constant increments in MD with the elevation of VO_2_ conductivity were perceived, and the maximum MD values of 97.03% and 69.04% eventually appeared at 0.362 THz and 0.485 THz, respectively. With the conductivity of Ge layers set to 4 × 10^3^ S/m, Figure 4c,d present MD varying with VO_2_ conductivity in the same frequency ranges, and the maximum MD values of 96.10% and 59.95% appear at 0.372 THz and 0.483 THz, respectively. The frequency shifts of maximum MD are caused by the coupling resonances between SRRs (contributed to by the conductivity of Ge layers), which produces specific influences on the transmission amplitude.

### 3.2. Simulation of Optically Tuned Ge Layers

The optical tuning properties of the proposed hybrid metasurface were investigated by simply varying the conductivity of Ge layers in the simulation, and the normalized transmission spectra without and with the electrical voltage excitation are depicted in Figure 5a,j. It was noticed that the transmission valley kept elevating with the increase in the conductivity of Ge layers. Independent of external stimuli, a transmission dip of 0.11 emerged at 0.565 THz, which then gradually rose to 0.469 as the conductivity of Ge layers was set to 4 × 10^3^ S/m. Identically, by fixing VO_2_ conductivity to 2 × 10^5^ S/m, a transmission dip of 0.285 occurred at 0.585 THz, which finally rose to 0.654 with the enhancement to the conductivity of Ge layers. Supposing VO_2_ conductivity of 2 × 10^2^ and 2 × 10^5^ S/m, changes in the transmitted H-field with the increase in Ge conductivity at 0.565 and 0.585 THz were observed, as shown in Figure 5b–i. Without optical pumping, the H-field was concentrated inside SRR_III_ and SRR_IV_, indicating the predominant position of magnetic dipole resonance in the transmission dip generation. As the Ge layer conductivity elevates, the gradual convergence of the magnetic field toward the electrodes and inside SRR_I_ and SRR_II_ manifests the dropped proportion of the magnetic dipole in the suppressed transmission. Meanwhile, the alternations in the H-field also disclose that other resonance modes are induced due to the interactions with SRR_III_, SRR_IV_, and nearby unit structures. Overall, the resonance strength of the whole unit tends to decline, resulting in improvements in the transmitted H-field and the consequent amelioration of transmission amplitude. 

The MD of the metasurface controlled by the optical stimulation was also calculated, as shown in Figure 6a,b. As the conductivity of Ge layers reached 4 × 10^3^ S/m, the maximum MD reached 77.08% at 0.565 THz and 56.36% at 0.585 THz for two different states of VO_2_ bridges. The reason for the difference can be interpreted by the changing resonance modes between the gold electrodes and SRR_I_ and SRR_II_, which in turn leads to disparate coupling strengths with the adjacent SRR_III_ and SRR_IV_.

### 3.3. Logic Operation Controlled by a Multifield

The proposed hybrid metasurface could find wide application. For instance, we further extended our design to realize the logic operation function. Two external input signals were configured as the voltage and optical pump, and the defined “0” and “1” input conditions with or without an external stimulus were respectively imitated by assigning the conductivity of VO_2_ bridges to 2 × 10^2^ S/m or 2 × 10^5^ S/m and Ge layers to 1 S/m or 4 × 10^3^ S/m. Notably, the output signals were contingent on weighing the normalized amplitude of transmitted THz waves through standardization of the transmission spectrum of the minimum valley in the frequency range. By exploiting the performance of the device controlled by the electrical and optical methods at 0.56–0.57 THz and 0.58–0.59 THz, the logical operation of two different gates can be independently realized, as depicted in Figure 7. 

The data in Figure 7a illustrate the “OR” logic gate executed by the metasurface. Here, the transmission spectrum when applying no external excitation (whereby VO_2_ conductivity was 2 × 10^2^ S/m and Ge conductivity was 1 S/m) was assumed to have the criterion value of 1. The other normalized amplitudes are observed to be more than four times greater in the range of 0.56–0.57 THz. Therefore, the output of amplitude can be considered as “0” in the absence of stimuli and “1” when modulated by either one or both of the approaches. Figure 7b illustrates the “AND” logic gate in the range of 0.58–0.59 THz, where the transmission spectrum was assumed to be the criterion at the VO_2_ conductivity of 2 × 10^5^ S/m and the Ge conductivity of 1 S/m. Except for the output tuned by both methods, which was taken as “1” for its amplitude being more than double that of others, the remainders were taken as “0” (for simply one or no stimulus). 

From the above analysis, the basic but critical “OR” and “AND” logic operation gates can be implemented, which greatly expands the application of the device. These logic gates are intended not only for simple numerical calculations, but also for functional operations applicable to many scenarios, such as read–erase–write manipulation of information that needs a clear amplitude difference from “1” and “0” output. Considering the “AND” logic gate, the device can output a “1” by individually modulating either VO_2_ or Ge, thus supplying two degrees of output timing freedom. From the “OR” logic gate, both materials need to be excited together to output a “1”. However, due to the memory property of VO_2_, it is possible to stimulate Ge at a certain frequency during the phase transition after the excitation of VO_2_, thereby achieving a continuous ultrafast output of “1”. Therefore, the device we propose possesses some similar functions, such as time-controlled read–erase–write operations. It also has simple time-manipulated information encryption and decryption capabilities when information is being read or written, which undoubtedly enhances the capability of contemporary VO_2_- or Ge-based information processing.

## 4. Conclusions

In summary, we have theoretically demonstrated a VO_2_ and Ge hybrid metasurface that supports electrical and optical control methods individually or simultaneously. To reveal the working mechanism of the designed device, the surface current and transmitted H-field were simulated, which testifies to the dominant role of the toroidal and magnetic dipole resonance strength respectively occupied by the two kinds of material components arranged in the meta-atoms. The switch on-off behaviors and modulation capability at four different frequencies were then investigated. Considering the distinct properties of modulated materials and tuning methods, the four switches possess nonvolatile or temporary on-off speeds to capture two kinds of temporal degrees of freedom. Furthermore, upon active excitation by two stimuli, the device manifests the logical operation functions of “AND” or “OR” gates in two frequency bands, which was confirmed by weighing normalized transmission amplitude. By integrating VO_2_ and Ge in one device, we have attained a THz metasurface with performance far beyond that of devices equipped with only one single material. Looking forward, such a hybrid metasurface with multiple functions and modulatory degrees of freedom could hold potential for THz photonic and electric meta-device design, which may greatly facilitate applications in data storage, ultrahigh-speed wireless communication, and optic-electric digital processing.

## Figures and Tables

**Figure 1 nanomaterials-12-03765-f001:**
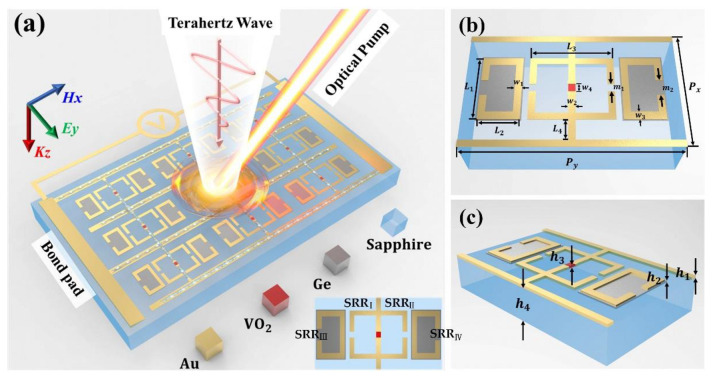
Schematics and working principle of the proposed hybrid metasurface. (**a**) Schematic illustration of the metasurface; the phase change material VO_2_ and the semiconductor material Ge are controlled by external voltage and the optical pump, respectively. (**b**,**c**) The detailed parameters of the unit cell.

**Figure 2 nanomaterials-12-03765-f002:**
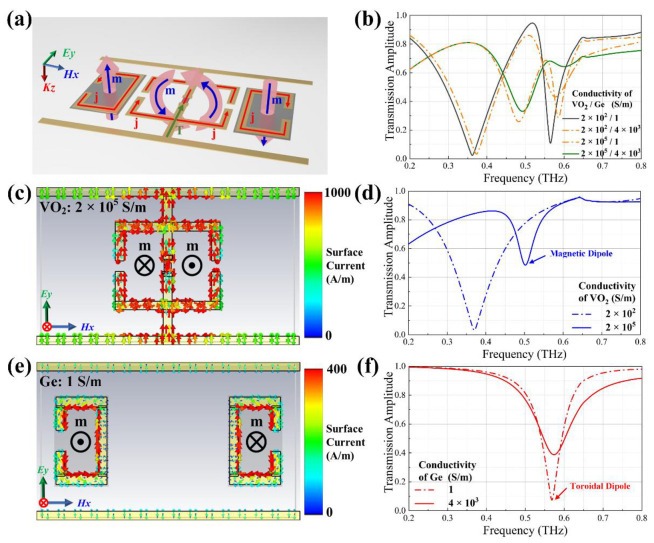
Resonance schematics and simulated transmission spectra of the designed unit. (**a**) Diagram of the transmit directions of the magnetic field and surface current. (**b**) Transmission spectra of the proposed metasurface for both materials with or without excitation. (**c**) Surface current distribution on SRR_I_ and SRR_II_ alone at the VO_2_ conductivity of 2 × 10^5^ S/m. (**d**) Transmission spectra of SRR_I_ and SRR_II_ alone for VO_2_ with or without excitation. (**e**) Surface current distribution on SRR_III_ and SRR_IV_ alone at the Ge conductivity of 4 × 10^3^ S/m. (**f**) Transmission spectra of SRR_III_ and SRR_IV_ alone for Ge with or without excitation.

**Figure 3 nanomaterials-12-03765-f003:**
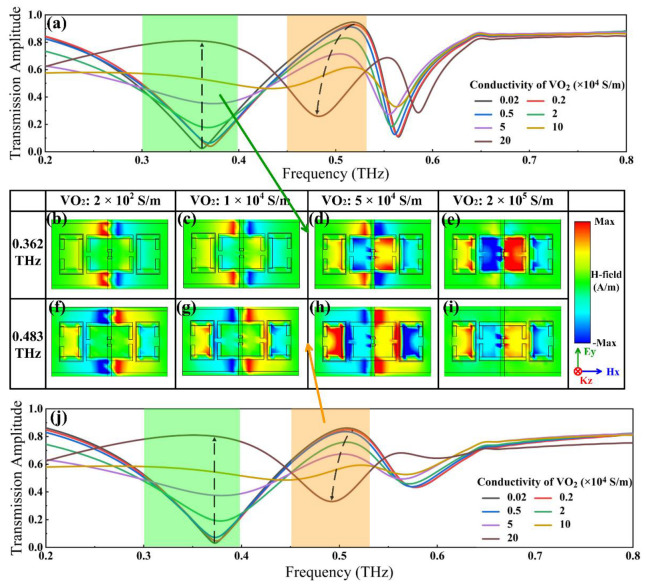
Characterizations of the hybrid metasurface modulated by the electrical method. Transmission spectra depending VO_2_ bridge conductivity when the conductivity of the Ge layer was fixed at (**a**) 1 S/m and (**j**) 4 × 10^3^ S/m. (**b**–**e**) Distributions of the H-field transmitted through the surface with the increase in VO_2_ bridge conductivity at 0.362 THz when fixing Ge layers at 1 S/m. (**f**–**i**) Distributions of the H-field transmitted through the surface with the increase in VO_2_ bridge conductivity at 0.362 THz when fixing Ge layers at 4 × 10^3^ S/m.

**Figure 4 nanomaterials-12-03765-f004:**
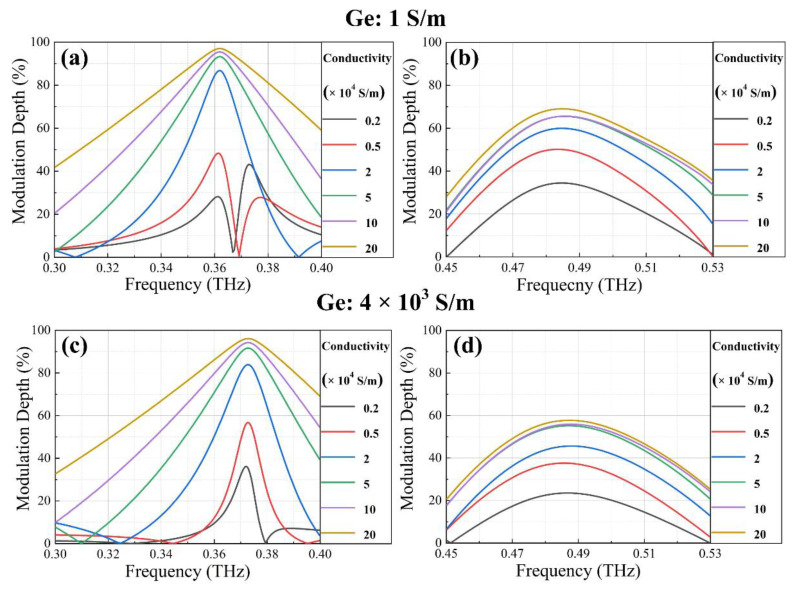
MD for varying VO_2_ conductivity with and without optical pump irradiation. (**a**,**b**) MD depending on the conductivity of VO_2_ bridges at 0.30–0.40 THz and 0.45–0.53 THz when the conductivity of Ge layers is 1 S/m. (**c**,**d**) MD depending on the conductivity of VO_2_ bridges at 0.30–0.40 THz and 0.45–0.53 THz when the conductivity of Ge layers is 4 × 10^3^ S/m.

**Figure 5 nanomaterials-12-03765-f005:**
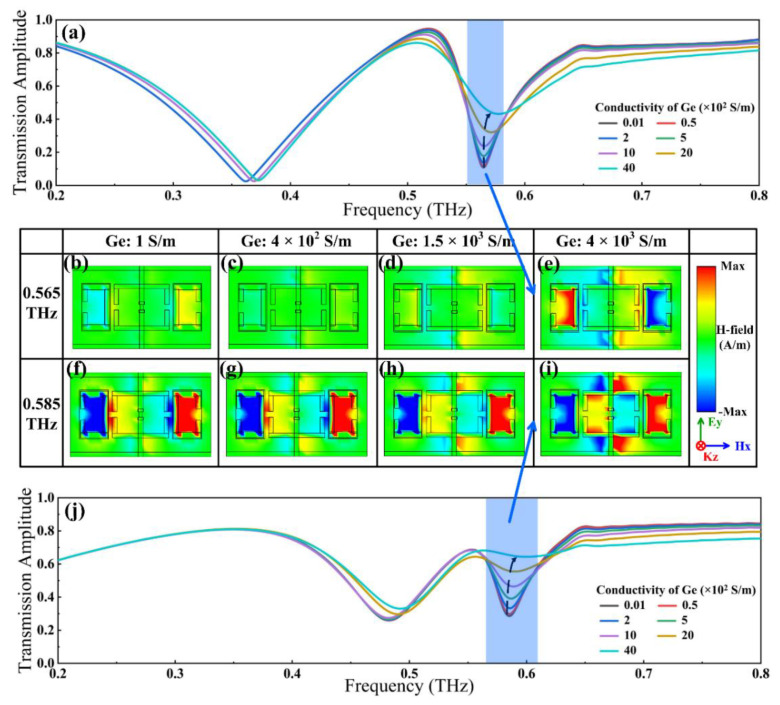
Characterizations of the hybrid metasurface modulated by the optical method. (**a**,**j**) Transmission spectra depending on Ge layer conductivity when conductivity of VO_2_ bridges is fixed at (**a**) 2 × 10^2^ S/m and (**j**) 2 × 10^5^ S/m. (**b**–**e**) Distributions of the H-field transmitted through the surface with the increase in Ge layer conductivity at 0.565 THz when fixing VO_2_ bridges at 2 × 10^2^ S/m. (**f**–**i**) Distributions of the H-field transmitted through the surface with the increase in Ge layer conductivity at 0.585 THz when fixing VO_2_ bridges at 2 × 10^5^ S/m.

**Figure 6 nanomaterials-12-03765-f006:**
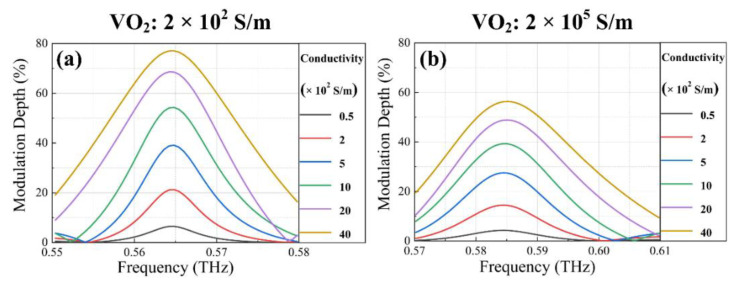
MD for varying Ge conductivity without and with the electrical voltage excitation. (**a**) MD depending on the conductivity of Ge layers at 0.55–0.58 THz when the conductivity of VO_2_ bridges is 2 × 10^2^ S/m. (**b**) MD depending on the conductivity of Ge layers at 0.57–0.61 THz when the conductivity of VO_2_ bridges is 2 × 10^5^ S/m.

**Figure 7 nanomaterials-12-03765-f007:**
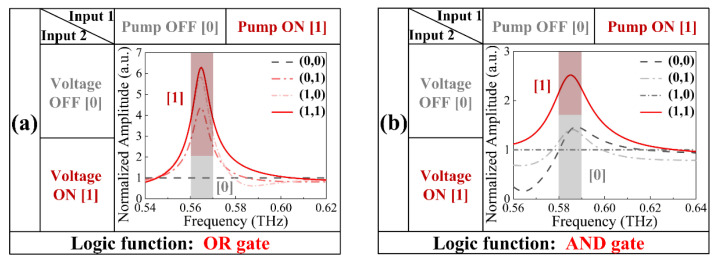
The dual-stimulus-controlled hybrid metasurface is functionalized as a logic (**a**) OR gate at 0.56–0.57 THz and (**b**) AND gate at 0.58–0.59 THz.

**Table 1 nanomaterials-12-03765-t001:** Parameter configuration of the proposed THz hybrid metasurface (μm).

** *P* ** ** _x_ **	** *P* ** ** _y_ **	** *L* ** ** _1_ **	** *L* ** ** _2_ **	** *L* ** ** _3_ **	** *L* ** ** _4_ **	** *w* ** ** _1_ **	** *w* ** ** _2_ **
150	90	50	29	60	15	5	5
** *w* ** ** _3_ **	** *w* ** ** _4_ **	** *m* ** ** _1_ **	** *m* ** ** _2_ **	** *h* ** ** _1_ **	** *h* ** ** _2_ **	** *h* ** ** _3_ **	** *h* ** ** _4_ **
6	6	6	12	0.5	0.3	0.5	5 × 10^5^

## Data Availability

Not applicable.

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
