# Peer review of "Multifield-Controlled Terahertz Hybrid Metasurface for Switches and Logic Operations"

_nanomaterials, 2022, doi:10.3390/nano12213765_

Round 1
Reviewer 1 Report
The authors proposed multifield controlled hybrid metasurface for switches and logic operations at THz frequencies. The concept is interesting, well described and gives a real perspective for its implementation. However, a few points require additional discussion:
1. In the introduction, the authors listed several concepts of metamaterials for the THz range, but omitted quite important works, including: Applied Physics Letters 99.23 (2011): 231101; Liquid Crystals 42.4 (2015): 430-434; Materials 15.8 (2022): 2777.
2. Were the conductivity values selected based on the available experimental data? This is crucial in order to determine the functionality of the device.
3. Amplitude modulation occurs in a quite limited spectral range, which significantly limits the application possibilities of the device. How can this property be improved?
4. What is the anticipated dynamics of the device, i.e. the switching times of the effect?
Reviewer 2 Report
The paper presents a hybrid metasurface that integrates both VO2 and Ge materials and can therefore be stimulated by both electrical and optical pumping. However, only simulations are presented and this is absolutely unacceptable when such high impact journal is targeted. Moreover, the novelty aspect is not shown here. Individually, both tuning mechanism has been reported and experimentally validated in literature.
Moreover, English has to be thoroughly revised. I propose some corrections that can be done before submitting this simulation work to another lower IF journal:
1. The inset in Fig. 1a should be modified. A pale blue background should be used and grey parts for SRR III and SRR IV for a better correspondence with the MS image
2. Fig. 2d cannot be understand. In fact, 6 curves are needed:
- 2 curves for SRR I/II (with and without excitation of VO2)
- 2 curves for SRR III/IV (with and without excitation of Ge)
- 2 curves for the whole structure SRR I/II/III/IV (with and without both excitations)
Reviewer 3 Report
In the manuscript, the authors have proposed a hybrid THz metasurface consisting of vanadium dioxide (VO2) and germanium (Ge) that enables electrical and optical tuning methods independently and simultaneously. By selectively triggering the VO2 or Ge in the meta-atom, toroidal or magnetic dipoles can be accordingly excited, thereby enabling the amplitude modulation and the consequence logic gate. The idea is interesting, and the paper is well written. I would recommend its publication once the following comments have been properly addressed.
1. Although the manuscript only presents the simulation results, it is better to discuss the fabrication process.
2. The expression of VO2 in the manuscript should be consistent.
3. Fig. 7 needs to be improved, especially the font sizes.
4. Some references on dynamic metasurfaces can be cited, such as Advanced Optical Materials 6 (9), 1701204, 2018; Scientific reports 9 (1), 5454, 2019; Nature Communications 13, 1-7, 2022.
Reviewer 4 Report
In the article ‘Multifield Controlled Terahertz Hybrid Metasurface for 2 Switches and Logic Operations’, the author proposes a tunable THz metasurface controlled by two active materials. In their design, through selectively ‘tuning on’ or ‘tuning off’ one active material, the device can work in different states, and they said this device can be a candidate for the THz logic device.
In my view, the individual adjustment part can be a reasonable design, different active materials have their own response speeds, so a device with two active choices can adjust the response speed, and enable a multifunctional device. Although in my view, such a design does not need to be integrated, the interference between electromagnetics can affect the design and cause unnecessary problems. Spatially separated two separate pieces make it easier to do the same things.
However, my main doubt is how to understand a logic gate with two different response speeds. If the speed is different, you can only operate on the lower response speed. So is that a better design if you design two center units (for example) controlled by two individual circuits using VO2? BTW, this design with two different active materials must be more expensive when you fabricated it. If it don’t have an obvious advantage, why we need it? Another doubt: in your design for a ‘OR’ or ‘AND’ gate, you use a manually specified judgment position, it seems in this way any individual control can do the same things. You must add some descriptions about the quality of your logic gates. What are the advantages of such a setup (efficiency, contrast, or something else) compared to other designs, apart from the inherent speed advantage of an optic setup?
A small tip: in Fig. 4(b) and (d) [also in Fig. 6], please adjust the y-axis range to at least be consistent. These are a set of comparative data, please do not use different coordinate ranges!
To conclude, the author must respond to the question of necessity, then I can consider whether the article is worth publishing.
Round 2
Reviewer 2 Report
The responses given by the authors concerning my comments did not make me change my former 'Reject' decision. As I stated previously, this work, though sound, is not mature enough to merit publication in such high IF journal. The authors compare their results obtained from simulations to experimental ones.
Reviewer 4 Report
Thank you for your reply, I think my doubts have been answered, and I agree to the publication of this article.